# Portrayal of NLRP3 Inflammasome in Atherosclerosis: Current Knowledge and Therapeutic Targets

**DOI:** 10.3390/ijms24098162

**Published:** 2023-05-03

**Authors:** Daniela Maria Tanase, Emilia Valasciuc, Evelina Maria Gosav, Anca Ouatu, Oana Nicoleta Buliga-Finis, Mariana Floria, Minela Aida Maranduca, Ionela Lacramioara Serban

**Affiliations:** 1Department of Internal Medicine, “Grigore T. Popa” University of Medicine and Pharmacy, 700115 Iasi, Romania; tanasedm@gmail.com (D.M.T.); dr.emiliavalasciuc@gmail.com (E.V.); ank_mihailescu@yahoo.com (A.O.); oana_finish@yahoo.com (O.N.B.-F.); 2Internal Medicine Clinic, “St. Spiridon” County Clinical Emergency Hospital Iasi, 700111 Iasi, Romania; minela.maranduca@umfiasi.ro; 3Department of Morpho-Functional Sciences II, Discipline of Physiology, “Grigore T. Popa” University of Medicine and Pharmacy, 700115 Iasi, Romania; ionela.serban@umfiasi.ro

**Keywords:** Inflammasome, atherosclerosis, NLRP3, IL-1β, IL-18, therapeutic target, NLRP3 inhibitors

## Abstract

We are witnessing the globalization of a specific type of arteriosclerosis with rising prevalence, incidence and an overall cardiovascular disease burden. Currently, atherosclerosis increasingly affects the younger generation as compared to previous decades. While early preventive medicine has seen improvements, research advances in laboratory and clinical investigation promise to provide us with novel diagnosis tools. Given the physio-pathological complexity and epigenetic patterns of atherosclerosis and the discovery of new molecules involved, the therapeutic field of atherosclerosis has room for substantial growth. Thus, the scientific community is currently investigating the role of nucleotide-binding and oligomerization domain-like receptor family pyrin domain-containing 3 (NLRP3) inflammasome, a crucial component of the innate immune system in different inflammatory disorders. NLRP3 is activated by distinct factors and numerous cellular and molecular events which trigger NLRP3 inflammasome assembly with subsequent cleavage of pro-interleukin (IL)-1β and pro-IL-18 pathways via caspase-1 activation, eliciting endothelial dysfunction, promotion of oxidative stress and the inflammation process of atherosclerosis. In this review, we introduce the basic cellular and molecular mechanisms of NLRP3 inflammasome activation and its role in atherosclerosis. We also emphasize its promising therapeutic pharmaceutical potential.

## 1. Introduction

Atherosclerosis, one of the 21st century’s fastest rising health emergencies, is defined as the accumulation of lipids, inflammatory cells, fibrous tissue, and calcification within vessels, especially in large arteries [1,2]. Exact data about its prevalence are hard to obtain, but its extent can be estimated by studying the multiple complications that result from and atherosclerotic disease [1,2]. The vascular pathological consequences of the macrovascular and microvascular systems, such as cardiovascular disease (CVD) and cerebrovascular events, are the most significant causes of morbidity and mortality in patients and place a substantial financial burden on the provision of equal access to treatment [3,4,5,6]. Of the four stages within the pathophysiology of atherosclerosis, comprising endothelial dysfunction succeeded by lipoprotein deposition, foam cell formation, inflammatory cell proliferation and migration, the inflammatory response appears to have the most prominent role. It is involved in both the initiation and the progression of atherogenesis [3,7]. After dyslipidemia emerged as a major risk factor for atherosclerosis which may lead to CVD, further research proved that atherosclerosis-related cardiovascular events are not entirely contingent solely on plasma lipid levels, prompting the emergence of additional risk factors underlying the development of atherosclerosis [3,6]. From the first reports about the inflammatory theory of atherosclerosis in 1999 [8] to the present date, more research is increasingly focused on systemic inflammation as a particular risk factor based on the relationship established between increased cardiovascular events and inflammatory markers, such as interleukin (IL)-6 and high-sensitivity C-reactive protein (hsCRP) [9]. Recent results from the Canakinumab Anti-inflammatory Thrombosis Outcome Study (CANTOS), in which anti-IL-1β treatment significantly reduced cardiovascular events independent of lipid levels, further support the idea of addressing inflammation as a key process in arresting atherosclerosis with a focus on inflammasomes [2,10,11].

There is increasing evidence that damage-associated molecular patterns (DAMPs), alongside hyperglycemia and hyperlipidemia, are associated with the accelerated onset of atherosclerosis via NLRP3 inflammasomes [12,13]. Furthermore, it has been established that NLRP3 inflammasome associated with the activation of interleukin-1β (IL-1β) and interleukin-18 (IL-18) amplifies vascular endothelial cell (VECs) damage, monocyte adhesion and infiltration, vascular smooth muscle cell (VSMC) proliferation, and promotes secondary plaque vulnerability [14,15,16]. 

While the role of inflammation in atherosclerosis is currently extensively explored, the precise deleterious effects on endothelial integrity and the involvement of molecules, such as NLRP3 inflammasomes, in the pathogenesis and evolution of atherosclerosis remain elusive [17,18].

In this narrative review, we aim to give an up-to-date perspective on the implications of NLRP3 inflammasome in atherosclerosis by describing their intricate physio-pathological relationship, their known therapeutic pathways and the existence of newer molecules that can modulate atherosclerosis via NLRP3 inflammasome. Finally, as a new approach, we point out in parallel the most recent reviews, their different focuses and their main highlights.

## 2. Portrayal of NLRP3 Inflammasome

As explained afterward, chronic inflammation is considered an essential part of the underlying multifactorial pathways of atherosclerosis, along with NLRP3 inflammasome activation. Inflammasomes were first introduced by Tschopp’s research in 2002 and soon after, innate immunity and cellular signal transduction became the main topics of research; inflammasomes in particular began to dominate this field of study [19,20].

Inflammasomes are intracellular protein complexes that are formed as pattern recognition receptors (PRRs) and interact either with DAMPs or with pathogen-associated molecular patterns (PAMPs) [21]. Examples of intrinsic molecules known as DAMPs that are generated in response to injury or distress include extracellular adenosine triphosphate (ATP) and cholesterol crystals (CCs). PAMPs are molecules of external origins, such as toxins generated by bacteria and viruses [22]. Although the number of other recognition receptors identified as having the ability to trigger inflammasome generation is consistently increasing, currently only nucleotide-binding oligomerization domain (NOD)-like receptor (NLR) family pyrin domain (PYD) containing 1 (NLRP1), NLRP3, NLR family caspase-recruitment domain (CARD) containing 4 (NLRC4), absent in melanoma 2 (AIM2) and pyrin are accepted as inflammasome receptors [6,14].

The NLRP3 inflammasome ubiquitously present in the cytosol of numerous cell types (monocytes, macrophages, T and B cells, fibroblasts) is the most extensively researched and prominent family member of inflammasomes [23]. Infections, cholesterol crystals, uric acid, bacteria, and a plethora of different ligands associated with underlying sterile inflammation in pathologies such as diabetes, hypertension, and atherosclerosis are just a few of the signals that cause NLPR3 to be activated [24,25,26]. By promoting caspase-1 activation to further break down pro-IL-1β and pro-IL-18 into mature and physiologically active forms (IL-1 and IL-18), it functions as a molecular switch for the inflammatory pathway that initiates and propagates atherogenesis [6,21]. 

Nevertheless, the precise pathway through which the NLRP3 inflammasome impacts atherosclerosis is elusive; hence, comprehending the inflammasome activation pathways is pivotal for developing innovative targeted and efficient treatments [27].

### 2.1. Structure of NLRP3 Inflammasome

The sequentially organized process of assembling an inflammasome usually involves a sensor protein, an adaptor protein, and an effector protein [12]. The NLRP3 inflammasome is a three-domain cytosolic protein compound consisting of three domains: a C-terminal leucine-rich repeat (LRR) domain, a central nucleotide-binding and oligomerization (NACHT/NOD) domain and an N-terminal effector PYD that interfaces with apoptosis-associated speck-like protein containing a caspase recruitment domain (ASC) [6,21,28].

The bridge between NLRP3 and caspase-1 is in turn mediated by ASC, an adaptor protein with an N-terminal PYD and a C-terminal caspase recruitment domain (CARD) [16,29]. In addition, the interaction with NLRP3 and other inflammasomal proteins, as well as ASC self-association, relies on the PYD domain [16,29]. Caspase-1, also known as the IL-1β converting enzyme (ICE), originally produced as an inactive zymogen through proteolytic cleavage [30,31], plays a major part in inflammation by mediating the conversion of the proinflammatory cytokines pro-IL-1β and pro-IL-18 into their mature and metabolically active forms, i.e., IL-1 and IL-18 [6,30,31]. 

In response to a particular stimulus, the NLRP3 sensing protein couples with ASC using homotypic PYD-PYD domain interactions, creating a single ASC “speck” residing within the activated cell and subsequently attracting pro-caspase-1 via CARD-CARD domain interactions [22,32]. Pro-caspase-1 undergoes autoproteolytic cleavage subsequent to the assembly of NLRP3, ASC and pro-caspase-1, releasing its active p20/10 subunits, which contribute to the self-inhibition of proteolysis. Additionally, proinflammatory cytokines from the IL-1 family, including IL-1β and IL-18, are cleaved by active caspase-1, releasing their mature forms [22,31,32,33,34]. Similar to the aforementioned, mature caspase-1 has just demonstrated its contribution to the proteolytic cleavage of gasdermin D (GSDMD), taking part in the development of the oligomeric membrane pore and inflammatory cell death mediated by inflammasomes known as pyroptosis [34]. Notably, pyroptosis is considered a highly inflammatory form of lytic programmed cell death mediated by the gasdermin family of proteins. It is initiated upon intracellular danger signals and acts as a defense mechanism against infection by inducing pathological inflammation, accompanied by the activation of inflammasomes and the maturation of pro-inflammatory cytokines [34].

### 2.2. Mechanisms of NLRP3 Inflammasome Activation

To date, several NLRP3 inflammasome activation pathways have been established: the canonical pathway which comprises of a two-signal model involving priming (signal 1) and activation (signal 2); a non-canonical pathway that necessitates caspase-4/caspase-5, secretion of IL-1β and IL-18 which respond to a particular intracellular lipopolysaccharide (LPS)-induced infection of gram-negative bacteria; and the alternative pathway driven by toll-like receptor 2 (TLR2) or TLR4 signaling without implicating additional secondary activators [35,36,37]. We further described the canonical pathway in the paragraphs that follow, given that is the main culprit for underlying atherosclerosis. 

Two steps are required for canonical NLRP3 inflammasome activation: initiation (signal 1) and activation (signal 2). Both constitute simultaneous defense mechanisms that tightly regulate inflammatory cells (Figure 1) [37].

The priming process involves the recognition of PAMPs and DAMPs by PRRs (TLRs, TL-1Rs, and cytokine receptors) followed by the activation of nuclear factor-κB (NF-κB), resulting in transcriptional activation of NLRP3, pro-IL-1β, pro-IL-18 and structural protein shifts, such as ASC phosphorylation and de-ubiquitination of NLRP3 [16,38]. The activation step involves NLRP3 oligomerization and NLRP3, ASC, and pro-caspase-1 complex formation, prompting caspase-1 activation as well as IL-1β and IL-18 generation [39].

Mechanisms of NLRP3 inflammasome activation continue to be a matter of interest, despite being thoroughly explored. Ion fluxes (K^+^ efflux, Ca^2+^ influx, and Cl^-^ efflux), mitochondrial malfunction, reactive oxygen species (ROS) accumulation, cathepsin B release from unstable lysosomes and trans-Golgi decay are several of the hypotheses that have been put forward [14,16,40].

#### 2.2.1. Ionic Fluxes

Most of the NLRP3 activators, which include extracellular ATP, nigericin (K^+^ ionophore), and particulate matter, activate inflammasome assembly via K^+^ efflux; this pathway has been deemed to be the predominant route of activation [41]. Conflicting findings have emerged as the alternative NLRP3 inflammasome pathway does not require K^+^ efflux, whereas the caspase-11-mediated non-canonical inflammasome pathway does [14,16,40]. More recent research has revealed that pharmacological substances, such as GB111-NH2, imiquimod and CL097, can activate NLRP3 by bypassing potassium efflux, implying that this event is presently accepted as a required but not crucial step for inflammasome activation [42].

In addition, potassium efflux triggers Ca^2+^ independent phospholipase A2, enabling IL-1β formation. Further cellular pathways, such as Ca^2+^ mobilization via calcium-sensing receptor (CaSR), Na^+^ influx and Cl^-^ efflux via volume-regulated anion channel (VRAC) and chloride intracellular channels (CLICs), were also proposed to indirectly activate the inflammasome by modulating K+ efflux [6,16]. The intermediate phase of never-in-mitosis A-related kinase 7 (NEK7)-NLRP3 molecular complex assembly underlies the mechanism by which K^+^ efflux-mediated activation occurs, being crucial for subsequent inflammasome activation [22,25,43]. Nonetheless, the precise regulatory mechanisms of these cellular processes are incompletely unraveled and still under dispute.

#### 2.2.2. Oxidative Stress

Another crucial element in NLRP3 inflammasome activation is the generation of mitochondrial ROS (mtROS). In particular, numerous studies have shown that ROS transform mitochondrial DNA (mtDNA) generated in response to NLRP3 activators into an oxidized form, thereby promoting inflammasome activation [6,16,44]. Whereas mtDNA interacts with NLRP3 and AIM2, the oxidized mtDNA is precisely required for triggering TLR signaling and NLRP3 inflammasome activation [16,44,45].

The manner in which NADPH-oxidase (NOX), another known important ROS generator, affects NLRP3 inflammasome activation is questionable. Although it was previously stated that NLRP3 inflammasome activation was not impacted by genetic and pharmaceutical inhibition of NOX, it has been proposed that NOX4 may play a role by controlling carnitine palmitoyl transferase 1A (CPT1A) and by increasing fatty acid oxidation, which is a potent inflammasome promoter [27,46]. ROS production in relation to oxidative stress also promotes inflammasome activation by dissociation of a thioredoxin-interacting protein (TXNIP) [27,46]. 

#### 2.2.3. Lysosomal Damage, Autophagy and the Trans-Golgi Network

Cathepsin release from the injured lysosome is an additional cellular driver of NLRP3 inflammasome activation. Degradation occurs if monosodium urate, CCs, asbestos, silica, β-amyloid, calcium crystals and silica particles are scavenged by macrophages but insufficiently degraded in lysosomes [40,47]. However, it is not yet known how lysosomal disruption and NLRP3 inflammasome activation are linked.

The build-up of diacylglycerol (DAG) was also linked to the potential involvement of the Golgi apparatus in NLRP3 inflammasome activation since Golgi membranes are surrounded by DAG and mitochondria-associated endoplasmic reticulum membranes (MAM) [43,48]. In response to NLRP3 activators, the protein kinase D (PKD) attraction induces NLRP3 phosphorylation, thereby facilitating inflammasome assembly [43,48]. The discovery of Nek7, alongside new developments in the role of a dispersed trans-Golgi network and mtDNA in NLRP3 inflammasome activation, constitutes important novel findings in this area [49].

It has been documented that autophagy is an alternative pathway that diminishes NLRP3 inflammasome activation through the removal of activators and intracellular components. In addition, pyrin, often referred to as the tripartite motif 20, serves as a targeted receptor that mediates precise autophagy of NLRP3 and pro-caspase-1 to trigger autodegradation [22]. Further research is warranted to unravel the activation process and the integration of stimulus-induced signaling events that activate the NLRP3 inflammasome [32].

#### 2.2.4. Regulation of NLRP3 Inflammasome

In addition, various other regulatory mechanisms, including post-translational modification (PTM), microRNA (miRNA) and endogenous modulator (CARD proteins, pirin proteins), control NLRP3 inflammasome expression and function. PTMs involving ubiquitination, phosphorylation, nitrosylation, sumoylation, glycosylation, and acetylation may control the initial activation of NLRP3 as well as its consequent priming [16,21,50]. In the regulation of ATS-NLRP3 inflammasome activation, several miRNAs, including miR-9, miR-155, miR-30c-5p, miR-181a, miR-181b-5p and miR-20a, were revealed to be implicated [43]. However, so far, this domain has not identified as a commonly shared mechanism of NLRP3 inflammasome activation. NLRP3 stimuli have been reported to induce activation of the inflammasome and a variety of cell signaling sequences. Arguably, the most critical signaling event amongst these is still the K^+^ efflux, which is required for most stimuli to activate the NLRP3 inflammasome [21]. The relevance of other events, including Ca^2+^ mobilization, Cl^-^ efflux, ROS, and mitochondrial dysfunction, are currently uncertain and under investigation [29,32,50]. 

### 2.3. Role of NLRP3 Inflammasome in Atherosclerosis

Localized inflammation in the vascular wall is prompted by dyslipidemia, high low-density lipoprotein (LDL) cholesterol and lipoproteins, which are all amplified in type 2 diabetes mellitus and enhance the development of atherosclerotic plaques [24]. IL-1β and IL-18, both by-products of NLRP3 inflammasome activation, appear to have a relevant contribution in the occurrence and propagation of atherosclerosis which is corroborated by abundant data obtained from the evaluation of atherosclerotic plaques in rodents and humans [26,31]. Besides the expression of adhesion molecules, such as the intercellular adhesion molecule-1 (ICAM-1) and the vascular cell adhesion molecule-1 (VCAM-1), inflammatory cytokines and chemokines, such as IL-6, IL-8, IL-1β, monocyte chemoattractant protein-1/chemokine(C-C motif) ligand 2 (MCP-1/CCL2) and matrix metalloproteinases (MMPs), elicit an inflammatory phenotype in endothelial cells and VSMC that enable macrophage build up [51]. While IL-18 receptors α/β are expressed in macrophages, endothelial cells and VSMC, IL-18 is expressed only in macrophages [40].

The discovery by Duewell et al. [52] in 2010 that low-density lipoprotein receptor (LDLR)^−/−^ atheroprone mice exhibit diminished atherosclerotic lesions consequent to their lack of NLRP3, ASC or IL-1α/β in bone marrow cells yielded the first concrete evidence that the NLRP3 inflammasome contributes to the onset of atherosclerosis [48,52]. This discovery paved the path for further research into this theory, and abundant examples of NLRP3 activating stimuli related to atherosclerosis are shown (Table 1); CCs are reported among the most effective activators of the NLRP3 inflammasome that occur during all phases of ATS [50,53]. Secondary to failure of macrophages to adequately achieve CC phagocytosis, it causes lysosomal instability and cathepsin efflux, which activates the NLRP3 inflammasome [50,53]. Moreover, it has also been documented that the generation of neutrophil extracellular traps (NETs), which primes macrophages, is initiated by CCs [21,50]. Furthermore, CCs contribute to a vicious cycle by enhancing NET release and macrophage priming, further complementing IL-1β, IL-18 and NET formation and NLRP3 inflammasome activation [6,50]. Nonetheless, the underlying pathway relating NETs to NLRP3 activation has not been fully determined and requires further exploration. Since oxidized low-density-lipoproteins (oxLDLs) can generate the initiation signal, this may be sufficient to send the NLRP3 inflammasome to both the activation and the priming signals [53]. The uptake of oxLDL by macrophage scavenger receptors, such as cluster of differentiation (CD) 36, initiates the formation of a TLR4/TLR6 heterodimer, which further activates the inflammasome and triggers NF-κB signaling [36,54].

ATP-dependent NLRP3 activation revealed itself as an important player in diet-induced atherosclerotic lesions via the purinergic 2X7 receptor (P2X7R), whose deficiency has been reported to suppress the extent of atherosclerotic plaques and decrease inflammasome activation [23,27,55].

Novel experimental data using animal and cellular designs of atherosclerosis provide mechanism-based perspectives on inflammasome modulation, particularly in the matter of diabetic macrovascular dysfunction [56]. Both in vitro and in vivo, NLRP3 promoted hyperglycemia-induced endothelial inflammation [13,57,58]. OxLDLs and high mobility group box protein 1, two agonist ligands of receptor for advances glycation endproducts (RAGE), have also been linked to NLRP3 activation which occurs in parallel with the atherosclerotic process in conjunction with hyperglycemia-induced ROS overproduction. Despite inconsistent published results, it is becoming evident that TXNIP, another redox signaling regulator, is significantly increased in response to hyperglycemia and may act as a direct ligand of the NLRP3 inflammasome [29,33,59]. Whilst advanced glycation end products(AGEs) have undisputedly contributed to diabetic atherosclerosis, it is not currently established whether the AGE/RAGE axis likewise activates NLRP3 in the atherogenic process [60,61,62]. Besides glucose toxicity, another potential inflammasome regulatory molecule, i.e., sterol regulatory element binding protein-1 (SREBP-1), is reported to be a key player in oxLDL-induced excessive lipid accumulation, causing foam cell formation and de novo lipid synthesis through the ROS-mediated NLRP3/IL-1β/SREBP-1 pathway [20,33,63].

Although endogenous factors that can activate inflammasomes (Figure 2) have been uncovered, additional research is required to comprehend how these signals are direct determinants of diabetes-related macrovascular dysfunction [23,64,65]. While endogenous signals that can activate inflammasome assembly have been identified, further research is needed to understand how they are directly linked to diabetes-related macrovascular injury. 

**Table 1 ijms-24-08162-t001:** Role of NLRP3 inflammasome in atherosclerosis.

Subjects	Notable NLRP3 Effects in ATS	Refs.
Ascending aortic tissue (CABG patients)	- NLRP3 expression higher in patients with AS and correlated with the degree of coronary artery disease	[66]
Human carotid atherosclerotic plaques	- NLRP3 inflammasome and components (ASC, caspase-1, IL-1β and IL-18) higher expression in unstable atherosclerotic plaques	[67]
Atherosclerotic plaques (ischemic cerebrovascular disease, MI patients)	- NLRP3–mRNA expression higher in symptomatic AS patients	[68]
Peripheral blood monocyte (chronic heart disease and acute coronary syndrome patients)	- NLRP3 inflammasome positive correlation with coronary atherosclerosis	[69]
ApoE^−/−^ mice fed with a HF and HM diet	- Increased NLRP3 expression and proinflammatory effect in hyperhomocysteinemia-induced atherosclerosis	[70]
ApoE^−/−^ mice fed with a HF diet	- NLRP3 inflammasome inhibition increased plaque stability	[71]
	- NLRP3 inflammasome inhibition reduced the size of atherosclerotic plaques and IL-1β and IL-18 levels	[72]
	- CCs activate NLRP3 inflammasome	[52]
ApoE^−/−^ mice chow diet	- NLRP3 inflammasome activation via Sirt3/FOXO3a/Parkin signaling pathway reduced atherosclerotic progression	[73]
ApoE^−/−^ mice western-type diet	- Specific NLRP3 inflammasome inhibition reduced atherosclerotic plaque development	[74]
Ldlr^−/−^ mice fed with PUFAs diet	- NLRP3 inflammasome inhibition reduced atherosclerosis by macrophage autophagy activation	[75]
ApoE^−/−^ mice	- The oxLDLs promote direct NLRP3 inflammasome activation and indirect via ERK1/2 pathway	[76]
ApoE^−/−^/caspase-1^−/−^ double knockout mice	- The extent of the area of atherosclerotic plaque reduced in caspase-1 deficient mice	[77]
Macrophages incubated with oxLDLs	- NLRP3 inflammasome activation, increase in IL-1β and IL-18 levels	[52]
NLRP3-deficient THP-1 cells	- NLRP3 inhibition reduces foam cell formation of THP-1 macrophages by oxLDL uptake suppression and increasing cholesterol efflux	[73]
HAECs	- NLRP3 inflammasome is activated by nicotine which promotes pyroptosis, proinflammatory cytokines secretion and atherosclerosis	[78]
Human and mice aortic endothelial cells	- Melatonin inhibits pyroptosis through the MEG3/miR-223/ NLRP3 signaling axis	[79]
HAECs	- NLRP3 inhibitor Microrna-30c-5p inhibits inflammation and pyroptosis via F0X03 pathway	[80]
VSMC	- AIM2 can stimulate caspase-1 via NLRP3 pathway and then mediates the inflammatory response by slicing GSDMD	[81]

coronary artery bypass grafting (CABG); myocardial infarction (MI); microRNA (mRNA); apolipoprotein E-deficient (ApoE^−/−^); high-fat and high-methionine (HF and HM); silent information regulator 3 (Sirt3); forkhead box O3 (FOXO3); LDL receptor knock-out (Ldlr^−/−^); polyunsaturated fatty acids (PUFAs); human aortic endothelial cells (HAECs); vascular smooth muscle cells (VSMC); absent in melanoma 2 (AIM2).

## 3. Atherosclerosis

Despite all the major advances in addressing risk factors, such as obesity, hypertension, and dyslipidemia, atherosclerotic-CVD manifested by myocardial infarction, stroke, and peripheral vascular disease, with an accelerated progression and concurrent impairment of several arterial territories, remains a primary cause of mortality with a significant negative influence on the quality of life [60,82]. Therefore, developing new strategies for atherosclerosis prevention and treatment is essential, given the rising prevalence and the severity of atherosclerosis-associated complications encumbered by its progression [83].

Endothelial dysfunction caused by oxidative stress, glyco-oxidation, and systemic inflammation amplified by hyperglycemia and dyslipidemia, allows mainly LDL and lipoprotein(a) (Lp(a)) to infiltrate vascular walls and enhance the migration of inflammatory cells through the expression of leukocyte adhesion markers, such as E-selectin, P-selectin, ICAM-1 and VCAM-1 [84,85]. LDL oxidation and an inflammatory immune response mediated by T lymphocytes and monocytes are prolonged in plaque-like lesions composed of inflammatory cells trapped in the subendothelial zone [10]. OxLDLs engulfed by monocyte-derived macrophages form foam cells that cluster in the vascular intima and cause necrotic core formation, inflammation, and phenotypic transition of VSMCs [60]. 

An extracellular matrix is produced in the fibrous cap as a result of VSMC migration and proliferation, collagen build-up and subsequent calcification [86]. As a result, the intima thickens and a self-perpetuating cycle of localized inflammation and apoptosis begins, causing gradual endothelial damage and the development of lipid-rich plaques with fibrous capsules, especially at the arterial emergence sites or bifurcations that are predisposed to blood flow alteration [87,88].

Whilst the early stages of endothelial dysfunction are relatively understood, it remains puzzling to comprehend how the progression and destabilization phases of atheromatous plaque take place [89].

### 3.1. Endothelial Dysfunction and Oxidative Stress

The majority of pathologies that are related to atherosclerosis display vascular endothelial dysfunction [90]. Local hemodynamics of blood flow constitute a regional risk factor for atherogenesis by producing injury prone areas primarily where laminar flow is altered. Endothelial cells, through their ability to modulate an optimal hemodynamic response to fluctuations in blood flow, play an instrumental role in atherogenesis [83,91,92]. 

A vicious loop that promotes apoptosis and increases extracellular matrix synthesis results in increased vascular permeability, activation of NOX, and worsened vasodilation due to decreased NO generation and increased ROS formation [88,93]. ROS production through the activation of several enzymes, including those in the mitochondrial respiratory chain, NOX, endothelial uncoupled nitric oxide oxidase (eNOS), cyclooxygenase, and xanthine (XO), provoke vasodilation abnormalities. This increases vascular permeability, thus encouraging the production of more adhesion molecules, such as ICAM-1, VCAM-1, and growth factors, such as vascular endothelial growth factor (VEGF), ET-1 and plasminogen activator inhibitor 1 (PAI-1), which accelerate the development of vascular sclerosis [94,95,96]. Increased activity of NF-κB triggers the production of proinflammatory cytokines, including IL-1β, IL-6 and TNF-α, and is secondary to reduced NO bioavailability, thus further attracting the creation of a pro-coagulant state by promoting the expression of tissue factor PAI-1 and the von Willebrand factor besides their role in maintaining the inflammatory milieu [60,86,95,97].

Under the influence of ROS production, as illustrated above, the structural integrity of the vascular endothelium is altered, resulting in the elevated expression of adhesion molecules (ICAM-1 and VCAM-1) and the adhesion of monocytes into the subendothelial space [2,10,98]. The differentiation of monocytes into macrophages attracts the release of IL-1β, IL-18, TNF-α, INF-γ, and MCP-1, leading to a vicious pathway of ROS production [88,93]. Subsequently, LDL cholesterol infiltrates into the subendothelial space in the intima, where it is deposited and modified into oxLDLs [99]. Further, the differentiation of monocytes in the subendothelial region into macrophages that release proinflammatory cytokines and ingest oxLDLs leads to the production of foam cells [97,99,100]. In addition, oxLDL causes the upregulation and release of inflammatory modulators, which further encourage the migration of monocytes, an increase in the density of scavenger macrophage receptors and the uptake of oxLDL during foam cell formation [99,101].

Furthermore, ROS potentiate various inflammatory pathways through redox modification of inflammatory mediators, such as DAMPs, transcription factors, nuclear factor erythroid 2-related factor 2 (Nrf2), NF-κB, hypoxia-inducible factor 1 (HIF-1) and activator protein 1 (AP-1), and through the formation of redox-dependent protein complexes (Nrf2- kelch-like ECH-associated protein 1 (Keap1)) [94,102]. In conjunction with causing insulin resistance, stimulation of the MAPK pathway also affects the phosphatidylinositol-3-kinase/serine-threonine kinase (PI3K/AKT) eNOS modulatory mechanism [86,98,103]. 

By modulating the NLRP3 inflammasome, the interaction between oxidative stress (OS) and inflammation through cytokine release is likewise documented [57,60]. As a result of a myriad of conditions, secondary dysfunctional endothelium implies a change from a quiescent to a proinflammatory phenotype and serves as the first stage in the development of an atherosclerotic lesion.

### 3.2. Inflammation in Atherosclerosis

As stated earlier in the previous part of this review, inflammation is a pivotal element in the pathophysiology of accelerated ATS, from the earliest stages of development to the final thrombotic outcomes; thus, it has a potential therapeutical endpoint [104]. The progression of atherosclerotic plaques is a continuum of processes involving both immune and non-immune vascular cells [10,24].

LDLs build up in the subendothelial region within the early stages of atherosclerosis, where they are transformed into oxLDLs and can activate ECs and macrophages in a proinflammatory manner, aggravate endothelial damage and attract leukocytes [10,99]. When VSMCs are subjected to modified LDLs, CCL2, CCL5, and MCP-1 are released, thus mediating monocyte recruitment [35,86,105]. In response to the local macrophage colony-stimulating factor (M-CSF)**,** underlying subendothelial monocytes divide into macrophages, which additionally release IL-1, IL-18, TNF-α, and INF- γ [24,35]. Under the influence of TNF-α, INF-γ and TLR ligands, macrophages adjacent to the lipid core differentiate into the proinflammatory M1 phenotype that correlates with self-reinforcing inflammatory processes, development of vulnerable plaques and the progression of atherosclerosis [35,106,107].

Another facet of atherosclerosis pathogenesis is the upregulation and activation of TLR4 and NLRP3 inflammasome and the nuclear transcription factor NF-kB as an extension of the endogenous host response to cholesterol efflux signaling misregulation, which is mediated by oxLDL and CCs [33,54,105,108,109]. Furthermore, NLRP3 inflammasomes have been linked to plaque development and progression through increased IL-1β and IL-18 production, increased MCP-1 and VCAM-1 and accumulation of the vascular extracellular matrix; these effects result from the JNK-induced apoptotic pathway, ASC and pro-caspase-1 [14,31,54,110,111]. In addition, the remodeling that proceeds with plaque development, intraplaque neovascularization, matrix depletion with thinning of the fibrous cap and eventually fibrous cap rupture causing thrombosis have all been connected to inflammation [2]. 

Considering the association between the accelerated progression of atherosclerosis induced by chronic sterile inflammation, addressing the disorder from this perspective may be a more reliable way to limit the development of its vascular consequences. Currently, no consensus guidelines are targeting specific inflammatory pathways that can improve outcomes for patients; therefore, there is a compelling need to further attractive research alternatives and therapeutic strategies that may arise from inhibiting inflammatory pathways. 

## 4. Therapeutic Targets

### 4.1. Hypoglycemic Agents

Anti-diabetic drugs have recently emerged as important novel mediators in treating diabetic vascular dysfunction and atherosclerosis by inhibiting NLRP3 inflammasome activation and reducing endothelial damage.

Glyburide, an FDA-approved sulfonylurea drug, interferes with the release of insulin from pancreatic β cells by inhibiting ATP-sensitive potassium channels (KATP) [57]. Glyburide was reported to exhibit anti-inflammatory effects mediated by the benzamide and sulfonyl group specific to NLRP3, thus blocking caspase-1 activation, IL-1β secretion and crystal-induced activation without relying on KATP channels and acting further downstream of P2X7R and upstream of inflammasome formation [112,113]. While glyburide effectively inhibits NLRP3 activation in vitro, the high doses required in vivo limit its usefulness as a treatment due to major adverse effects. Its exact mechanism of action remains not yet fully understood [21].

Since 1995, metformin, which was originally synthesized in 1922, has become broadly recognized as the first-line treatment for T2DM, thus starting a new phase in the growing burden of diabetes [114]. The main therapeutic mechanisms behind the drug are primarily the result of adenosine monophosphate-activated protein kinase (AMPK) activation, which further reduces hepatic gluconeogenesis and improves insulin resistance while improving glucose uptake in peripheral tissues [115]. Metformin is currently being investigated for new roles and pharmacological processes in light of growing interest in the role of inflammation in the etiology of both T2DM and atherosclerosis [114,116]. The main data revealed that metformin increased the production of AMPK and protein phosphatase 2A (PP2A) expression, leading to reduced expression and inhibition of NLRP3 inflammasome activation in oxLDL-stimulated macrophages [117]. These findings were additionally supported by the fact that metformin averted accelerated diabetic AS in apoE^−/−^ mice by activating the Trx-1/Txn pathway [32,116,118].

The Empagliflozin Cardiovascular Outcome Event Trial in Type 2 Diabetes Patients (EMPA-REG OUTCOMES) study reported that treatment with sodium–glucose cotransporter 2 (SGLT2) inhibitors, a novel class of hypoglycemic agents, reduced the rate of all-cause cardiovascular death in patients with T2DM at high risk of cardiovascular events [119]. The mechanism of action is mediated by the proximal kidney tubules, where SGLT2 inhibitors reduce renal glucose reabsorption and increase urinary glucose excretion [32,57]. In ex vivo research employing human macrophages, empagliflozin administration, in addition to previous effects, diminished NLRP3 inflammasome activation and IL-1 release, partly by increasing β-hydroxybutyrate (BHB) and by reducing serum insulin, glucose and uric acid levels [120].

Dapagliflozin, another family member, alleviated diabetic cardiomyopathy by inhibiting NLRP3 through activation of the AMPK system and blocked the TXNIP/NLRP3 pathway [54,121]. Furthermore, dapagliflozin therapy suppressed the generation of serum NLRP3, IL-1β, IL-18 levels and ROS in the vasculature of atherosclerotic aortic lesions, thus reducing the progression of ATS, diminishing macrophage infiltration and improving lesion stability [122]. Regardless, the reported results strongly indicate compelling reasons to further research the effects of SGLT2 inhibitors as a therapeutic solution for NLRP3 inflammasome inhibition in diabetic patients [32,57]. 

Apart from their hypoglycemic action, dipeptidyl peptidase 4 (DPP-4) inhibitors and glucagon-like peptide receptor (GLP-1R) agonists have been demonstrated to improve inflammatory markers, oxidative stress, endothelial function and, to some extent, the atheroprotective features in patients with T2DM [123]. Saxagliptin was reported to reduce myocardial injury by inhibiting the ERK/TLR4/NLRP3 and p38/miR-146b/TLR4/NLRP3 pathways, implying that DPP4 inhibition blocks the first step involved in inflammasome activation [124]. In oxLDL-induced THP-1 cells, sitagliptin elicited a substantial downregulation of NLRP3, TLR4 and IL-1 expression and an increase in GLP-1R expression [43]. Vildagliptin, as per Qi et al., mitigates endothelial dysfunction mediated by elevated free fatty acid (FFA) levels by inhibiting the AMPK-NLRP3-HMGB1 pathway, preserves mitochondrial function and restores eNOS levels while decreasing cellular lactate dehydrogenase (LDH) production and ROS levels [125]. Anagliptin, a new DPP-4 inhibitor approved for treating T2DM, was recently reported to reverse endothelial dysfunction by SIRT1-dependent inhibition of NLRP3 inflammasome activation and suppression of NOX4-ROS-TXNIP-NLRP3 crosstalk, in this way creating scope for additional research [46,126]. In addition, glucagon-like peptide-1 receptor (GLP-1 RA) agonists provide beneficial effects on the cardiac function as well. Dulaglutide, a novel representant of the class, inhibits NLRP3 inflammasome, NOX4 and TXNIP expression in endothelial cells in a SIRT1-dependent manner, protecting against the effects of increased glucose on the NLRP3 inflammasome activation [43,46,127].

### 4.2. Direct and Indirect NLRP3 Inhibitors

Aiming to prevent inflammasome-mediated cell death and thereby decrease local inflammation, pharmacological suppression of NLRP3 inflammasome activation may represent a more precise and certain therapeutic approach for CVD [21]. Agents with small molecules, such as MCC950, dapansutrile (OLT1177), CY-09 or tranilast, block NLRP3 inflammasome activation by specifically targeting the NATCH domain of the NLRP3 structure while numerous other inhibitors act by impairing ATPase function, such as 3,4-methylenedioxy-nitrostyrene (MNS), Bay 11-7082, BOT-4-one, parthenolides and INF39 [128]. Furthermore, because of many other biological functions, as shown in more detail below, these agents are unlikely to function as specific NLRP3 inhibitors [21]. Nevertheless, the scarcity of in vivo studies and NLRP3-dependent models underlines the shortage of information on some compounds and the reasons why they remain unapproved as treatments.

#### 4.2.1. MCC950

MCC950, a sulfonylurea compound that was first identified in 2001 as CP-456773 and subsequently known as cytokine release inhibitor drug 3 (CRID3), is an effective and selective NLRP3 inflammasome inhibitor that binds to a specific residue in the walker B motif of the NATCH domain of NLRP3 to prevent ASC oligomerization and NLRP3 ATP hydrolysis [129]. Since it was established as an NLRP3 inhibitor, it has been used as a pharmacological instrument to unravel the pathogenic functions of the NLRP3 inflammasome in several disease models, including T2DM, ATS, and many others [46]. Corcoran et al. comprehensively analyzed its functions in more than 100 preclinical models of inflammatory diseases [130]. At nanomolar doses, MCC950 inhibits NLRP3 inflammasome activation by suppressing IL-1β secretion but not NLRP1, NLRC4, or AIM2 inflammasomes, suggesting a capacity to modulate the immune system by only partial interleukin inhibition [35,59,131]. By indirectly influencing intraplaque macrophage composition or through processes such as macrophage proliferation, the diminished size of atherosclerotic lesions represents the probable therapeutic benefit of MCC950 in ApoE^−/−^ mice [43]. In addition, through sirtuin 1 and superoxide dismutase (SOD), tissular overexpression of MCC950 exhibited antioxidant capacity [132]. Regardless of the favorable outcomes in animal models, the compound is currently withheld from clinical trials because of its liver toxicity [24,36]. 

#### 4.2.2. CY-09

By interfering directly with the ATP-binding Walker A motif protein, another small-molecule inhibitor, CY-09, selectively inhibits the NLRP3 inflammasome ATPase activity [21,133]. Ex vivo and in vivo designs for cryopyrin-associated periodic syndromes (CAPS), type 2 diabetes, gout and non-alcoholic fatty liver disease (NAFLD) have established the efficacy of CY-09, although further research is required to materialize its full therapeutic value [134,135].

#### 4.2.3. OLT1177/Dapansutrile

OLT1177, also referred to as dapanustrile, is a nitrile sulfonyl derivative that selectively inhibits the inflammasome ATPase activity, disabling NLRP3-ASC and NLRP3-caspase-1 interactions and IL-1β generation in neutrophils and monocyte-derived macrophages [136]. In a phase I clinical trial, dapanustrile, proved to be safe and tolerated in oral application in healthy adults, with both in vitro and in vivo efficacy distinguishing it from MCC950 [36,137].

#### 4.2.4. Tranilast

Originally discovered to suppress NF-κB by disrupting the association between NF-κB and CREB-binding protein (CBP), tranilast, an analogue of a tryptophan metabolite, has been utilized to treat bronchial asthma in Japan, China and South Korea since 1982 [43]. Highlighted in current emerging research as a selective NLRP3 inhibitor and IL-1β down-regulator, in contrast to other direct inhibitors, the compound mediates only the NLRP3-ASC interplay and does not affect ATPase activity of the protein [138]. Tranilast suppresses inflammation by inhibiting NLRP3 inflammasomes but not AIM2 or NLRC4 inflammasomes. Additionally, it blocks LPS-induced pro-IL-1β and IL-6 production, despite having an in vivo potency that is 5–10 times less effective than MCC950 [21].

In addition to its anti-inflammatory properties, it further exhibited antioxidant effects demonstrated by ROS scavenging and direct suppression of xanthine oxidase activity in vitro, as well as reducing TXNIP expression and ROS generation in rats that were administered streptozotocin [36].

#### 4.2.5. Oridonin

Oridonin (Ori), a natural constituent derived from the plant *Rabdosia rubescens*, is frequently consumed in East Asia as a dietary supplement for its anti-inflammatory, antitumor, antimicrobial and neuroprotective effects. Through the disruption of NF-κB signaling, nuclear translocation and DNA binding, it mediates the suppression of various pro-inflammatory regulators [139,140,141]. Recently, ori has emerged as a selective, non-reversible inhibitor of NLRP3 that suppresses IL-1β generation and NLRP3-NEK7 interaction by covalently altering the C279 cysteine residue in the NACHT domain without impacting the ATPase [142]. 

Concerns regarding the toxicity and the detrimental pharmacological consequences of ori exist, despite the in vivo effects observed in preclinical investigations [143]. To increase pharmacological activity and bioavailability, more studies on ori analogues are warranted.

#### 4.2.6. 3,4-Methylenedioxy-β-Nitrostyrene (MNS)

By interfering with the synthesis of IL-1β, IL-18 and caspase-1 activation, the spleen tyrosine kinase (Syk) and Src tyrosine kinase inhibitor, 3,4-methylenedioxy-β-nitrostyrene (MNS) is likewise a reliable and selective inhibitor of the NLRP3 inflammasome [144]. Subsequent findings indicate MNS binds to the LRR and NACHT domains of the NLRP3 protein, inhibiting ATPase function but not the inflammasome AIM2 or NLRC4. However, its extreme toxicity hindered further research [47].

#### 4.2.7. INF Analogues

INF analogues were first introduced in 2014; one of the compounds, INF4E (compound 9), irreversibly inhibits the NLRP3 protein via its Michael acceptor. They display a potential role as targets for IL-1β secretion, ATPase and caspase-1 activity inhibition [145]. In addition, INF4E inhibits ATP- and nigericin-induced cell death and inhibits pyroptosis, demonstrating novel additional cardioprotective benefits in myocardial ischemia/reperfusion injury via the ERK/Akt/GSK-3β RISK pathway [54,146]. The discovery of the analogue INF58 (compound 14), which irreversibly inhibits the NLRP3 ATPase with enhanced potency, and INF39 (compound 11), which has reduced cytotoxicity and reactivity while attenuating NEK7-NLRP3 interactions, was made possible by derivatization and complementary structural analysis [145].

With no inhibition of AIM2 or NLRC4 inflammasomes but inhibition of ASC speck production and cleavage of caspase-1, IL-1β and GSDMD proteins, INF39 is the most studied representative as a potential future NLRP3 selective inhibitor [147].

#### 4.2.8. BAY 11-7082

BAY 11-7082 (BAY), a phenylvinylsulfone with multiple pharmacological and potential pathways, is known to reduce TNF-α, which inhibits the nuclear factor-κB (IκB) kinase (IKK) phosphorylation that in turn reduces NF-κB signaling and NLRP3 production [36,43]. By completely alkylating cysteine residues in the NLRP3-ATPase region through a Michael coupling process, BAY decreases NLRP3’s ATPase function and prevents ASC oligomerization [148]. The covalent cysteine residue C191 alkylation by the compound is involved in the GSDMD pore formation, as was observed by Hu et al. [149].

#### 4.2.9. VX-740 (Pralnacasan) and VX-765

The orally targeted caspase-1 inhibitors VX-740 (Pralnacasan) and their analogue VX-765 represent two novel compounds that covalently alter the catalytic cysteine residues in the active sites of caspase-1, thereby impairing caspase-1 cleavage and pro-IL-1β/18 processing [150]. While both rheumatoid arthritis (RA) and osteoarthritis (OA) have been successfully managed with VX-740 in phase IIa clinical trials, additional research has been suspended due to the severe hepatotoxicity uncovered with prolonged use [43]. Since VX-765 was proved to significantly downregulate pyroptosis and IL-1β expression in SMCs incubated with oxLDL and suppress AS formation in ApoE^−/−^ mice, it has become the most researched caspase-1 inhibitor [151].

#### 4.2.10. Anakinra (Kineret), Rilonacept (Arcalyst) and Canakinumab (Ilaris)

Anakinra is a synthetic IL-1Ra. Rilonacept is a dimeric fusion protein with decoy receptors containing extracellular residues of the two IL-1R subunits. Canakinumab is an anti-IL-1β monoclonal antibody that specifically binds to and neutralizes IL-1β. These are the three biological therapies currently approved by the FDA that additionally target the NLRP3 pathway [6]. Neither of the biologics previously listed has received approval for use in clinical practice as an NLRP3 inhibitor alone, despite the NLRP3-IL-1b inflammasome pathway being a potentially valuable therapeutic target for patients with atherosclerosis and other inflammatory conditions [15]. Regarding CVDs, some clinical trials with anakinra have been carried out in patients with myocardial infarction (VCU-ART and VCU-ART2; MRC-ILA-Heart Study) [152,153], but the stated outcomes are controversial due to the sparse patient samples involved [15,154].

A large-scale clinical trial, the Canakinumab Anti-inflammatory Thrombosis Outcomes Study (CANTOS), was conducted to explore the theory of inflammation interplay in ATS [11]. Although neutralizing IL-1β with the canakinumab antibody (at a dose of 150 mg every three months) improved cardiovascular events related to atherosclerosis, a major impediment in admitting this compound as a therapy are the undefined strict criteria for eligibility as the study revealed substantial side effects of the treatment [11,24,43]. 

#### 4.2.11. Colchicine

In addition to its proven proficiency in treating gouty arthritis and pericarditis, colchicine, a widespread drug, has proven recently to be an effective anti-inflammatory compound by blocking NLRP3 inflammasome activity and suppressing IL-1β and IL-18 secretion [31,43].

Patients with acute coronary syndrome who were administered short-term colchicine in the Colchicine Cardiovascular Outcomes Trial (COLCOT) displayed markedly decreased IL-1 and caspase-1 production when compared to untreated patients, indicating that colchicine has potential atheroprotective effects [155]. The incidence of cardiovascular mortality, myocardial infarction, ischemic stroke and ischemia-induced coronary revascularization was decreased by 31% in the Low-Dose Colchicine (LoDoCo20) trial, which enrolled patients with chronic stable CAD after one month of colchicine treatment (0.5 mg once daily). The LoDoCo2 trial that included patients with stable chronic CAD after one month of colchicine use (0.5 mg once daily) had a 31% reduction in the incidence of cardiovascular death, myocardial infarction, ischemic stroke and ischemia-driven coronary revascularization when compared with patients receiving placebo [156]. The results reinforce the promising advantages of anti-inflammatory agents in patients with coronary artery disease and are concordant with those of the COLCOT study and the first LoDoCo study [24,155,157].

Even though CANTOS and LoDoCo2 have still not adjusted the therapeutic agenda for cardiovascular risk reduction in clinical practice [11,156], these pivotal studies constitute an important step in the practical implementation of immunomodulatory therapies targeting NLRP3 for CVD.

#### 4.2.12. Less Known NLRP3 Inhibitors

Other NLRP3 inflammasome inhibitors are currently being studied, but they are in an early stage of research and, considering that new molecules are constantly emerging, require further evaluation to be considered as potential future therapies. 

The benzoxathiol derivative BOT-4-one exerts anti-inflammatory properties through various pathways, including inhibition of the NF-κB pathway by alkylation of IKKβ and consequently decreased production of NLRP3 [149]. Although the effect on ubiquitination is uncertain, the molecule’s capacity to arrest ATP-induced IL-1 release deserves further exploration in atherosclerosis [47,158]. By disrupting the transcription factors NF-κB and signal transducer and activator of transcription 1 (STAT1), methylene blue, an anti-inflammatory, antioxidant and neuroprotective substance, has lately proved its effects in reducing NLRP3, pro-IL-1β and iNOS expression [159,160].

Disulfiram, the FDA-approved treatment in alcohol dependence, additionally exhibited lysosomal protection, mitochondria-independent ROS generation and NLRP3 inhibition [161]. In particular, disulfiram has been reported to suppress GSDMD pore development, hindering IL-1 β release and associated pyroptosis [162]. Supplementary research is needed to retarget disulfiram treatment for inflammatory pathologies.

Some phenamic acid derivatives have been reported to interact with NLRP3 inflammasomes by disrupting the Cl^-^ volume-regulated anion channel (VRAC) as a complement to their extensive use as nonsteroidal anti-inflammatory drugs (NSAIDs) that inhibit cyclooxygenase (COX) [163]. The detection of pleiotropic effects in antidepressants has been enabled by acknowledgement of the importance of inflammatory routes in the pathophysiology of psychiatric diseases. Recently, fluoxetine revealed direct NLRP3 inhibitory effects and downstream IL-1β regulation in macrophage cells and retinal pigment epithelium (RPE) cells [164]. This opens potential new areas of use for fluoxetine and phenamic acid derivatives, but further studies are necessary to determine their efficacy in various conditions. Another innovative approach is ursodeoxycholic acid (UDCA), which decreased intracellular CC by dissolving in macrophages and thereby reducing IL-1β secretion [165].

Considering its entanglement in inflammatory disorders, with the direct consequence of endorsing cytokine release and mediation of endothelial pyroptosis, NLRP3 inflammasome is an attractive therapeutic target. As observed, new pharmacological approaches hamper its pathway by targeting a specific phase via different components or by directly inhibiting NLRP3 and fully stopping its effects (Table 2).

### 4.3. Statins

Modern research shows that statins, as analogues of 3-hydroxy-3-methylglutaryl coenzyme A inhibitor (HMG-CoA) and typically utilized for their cholesterol-lowering ability, also have pleiotropic effects, including immunomodulatory and anti-inflammatory abilities.

Wu et al. [170] revealed that atorvastatin, the most researched member of the class, inhibits the NLRP3 inflammasome and the molecules involved in pyroptosis, such as caspase-1 and IL-1 β, by increasing the production of NEXN-antisense RNA1 (AS1) lncRNA and its related gene NEXN [170,171]. In addition, the anti-inflammatory effects translated as IL-1β downregulation are supported by inhibition of the TLR4/MyD88/NF-κB pathway demonstrated in THP-1 cells stimulated with phorbol 12-myristate 13-acetate (PMA) [27,172]. In this account, it is reasonable to hypothesize that reduced NLRP3 inflammasome signaling is at least broadly accountable for the effects of atorvastatin on the progression of atherogenic pathways and chronic inflammation [43].

Simvastatin therapy also substantially decreased expression levels of NLRP3, cathepsin B, IL-1β and IL-18 in peripheral blood monocytes in response to CC stimulation both in vitro and in vivo [32,173]. Additionally, another atheroprotective pathway emerged and is exemplified by the endothelial Kruppel-like factor 2 (Klf2)-Forkhead box P transcription factor 1 (Foxp1)–NLRP3 inflammasome network; it warrants more consideration for its potential effect in delaying T2DM-associated atherosclerosis progression [46,174]. Another class representative, rosuvastatin, displayed in vivo decreased NLRP3 activation mostly via crosstalk with the oxidative stress enzymatic players, such as SOD, glutathione peroxidase and catalase (CAT) [54,175].

When considered collectively, preclinical research findings imply that statins have the potential to be an efficient therapeutic and prevention approach for cardiovascular disease by controlling NLRP3 activation. 

### 4.4. Natural Compounds

A vast array of novel substances identified from a variety of natural compounds that have been used for many years in conventional, traditional medicine qualify as potential NLRP3 inflammasome inhibitors that can provide effective therapies. However, the mechanisms underlying their immune modulatory function have not been thoroughly investigated.

#### 4.4.1. Flavonoids

Many fruits, vegetables and grains contain a phytonutrient class commonly known as flavonoids; it is widely recognized for its neuroprotective, anti-inflammatory, and antioxidant effects. The ability of flavonoids to decrease ROS production and expression of inflammasome components, such as ASC, NLRP3 and active caspase 1, as well as the consistent release of IL-1β, are presumed to be molecular mechanisms by which they modulate inflammasome activity [47,176]. The most promising NLRP3 inflammasome antagonists will hereafter be outlined. 

Reductions in levels of the pro-inflammatory cytokines IL-6, IL-1β and TNF-α, coupled with additional inhibition of the ERK1/2 and NF-κB pathways, are accomplished by the flavone apigenin, contained in vegetables such as celery and parsley [32,177]. Although apigenin does not affect ASC protein levels despite reduced NF-κB activation, it has recently been reported to reduce phosphorylation of two key enzymes involved in ASC phosphorylation: Syk and the protein tyrosine kinase 2 (Pyk2) [176]. Additional research is required to define the specific NLRP3 inhibitory mechanism since the compound exhibits non-selective function by interfering with IL-1β production via AIM2 inflammasome [178].

Cardamonin, a naturally occurring chalcone identified in the Alpinia katsumadai Hayata plant, has been documented to reduce nitric oxide (NO) levels via NF-κB inhibition and by increasing the AhR/Nrf2/NQO1 pathway, a recognized NLRP3 downregulator [179]. However, there is a lack of research papers on the effects of cardamonin on ATS [180].

Because of their high concentrations of flavonoids, chalcones and other phytonutrients, including isoliquiritigenin (ILG) and glycyrrhizin (GL), two extracts from the Glycyrrhiza uralensis (licorice) plant, have traditionally been utilized to treat a series of conditions [47]. Either compound inhibits NF-κB and MAPK activity by repressing the TLR4/MD-2 molecule [54]. A single study concluded that isoliquiritigenin therapy reduced NLRP3 inflammasome, caspase-1 activation and IL-1β generation in obesity through interference with both the inflammasome initiation and activation phases [181]. Albeit in a non-specific manner modulated by NF-κB luteolin present in plants such as broccoli, celery, thyme and pepper, these compounds can reduce ROS and NLRP3 generation besides other inflammatory mediators [182,183]. Further testing raised the hypothesis of luteonin inflammasome inhibitor potential, but none focused on ATS [182,184]. IL-1β secretion via NLRP3 and AIM2 inflammasome pathways is diminished by the non-selective inflammasome inhibitor quercetin, commonly present in a variety of grains, vegetables, and fruits [185]. Quercetin represses the NLRP3 inflammasome activation phase by impairing ASC-speck oligomerization, although the exact mechanism is currently unclear [47,186].

#### 4.4.2. Phenols

Several species of the genus *Artemisia* have prompted interest in further study of their anti-inflammatory benefits due to their current use in traditional Asian remedies. As a pertinent example, the dysregulated activation of NLRP3 or AIM2 inflammasomes yields the anti-inflammatory effects of Artemisia princeps extract (APO), which suppresses the generation of ASC speck [187,188]. Additional constituents, such as Artemisinin from *Artemisia annua*, ameliorated foam cell production by impairing the inflammatory AMPK/ NF-*κ*B-NLRP3 pathway response in macrophages [32,189]. By modulating ASC-PYD binding and inhibiting NLRP3-ASC interplay, the phenolic class representant caffeic acid phenethyl ester (CAPE) found in bee propolis might decrease NF-*κ*B activation and hinder NLRP3 and AIM2 inflammasome activation [190].

Curcumin, a common polyphenol substance renowned for its anti-inflammatory and antioxidant properties, reduced NLRP3 expression due to IKK phosphorylation inhibition that further impaired the TLR4/MyD88, NF-κB and P2X7R inflammasome pathways [32,191], thus dismissing the hypothesis of inhibition by the antioxidant mechanism [192]. 

While a promising target for suppressing the AIM2 inflammasome, the terpene group compounds, such as andrographolide and parthenolide, produce only mild inhibitory effects on the NLRP3 inflammasome [47,192,193].

#### 4.4.3. Miscellaneous

The traditional Chinese herbal naftoquinone compound Shikonin, extracted from the roots of *Lithospermum erythrorhizon*, displays numerous anti-inflammatory mechanisms by targeting NF-κB suppression and NLRP3 mediated IL-1β production; however, the effects on cardiovascular disease are still unknown [194,195]. In addition, *Salvia miltiorrhiza* Bunge contains salvianolic acid A (SAA), a phenolic molecule that hinders NLRP3 activation in aortic tissues via the NF-κB pathway and reduces early-stage ATS [196]. Moreover, 6-shogal, the most potent ginger-derived compound, displayed positive effects by reducing Akt activation, ROS production and NLRP3 inflammasome activation, thereby alleviating hyperglycemia-induced calcification of arterial smooth muscle cells [55,197]. Inhibition of ROS/TXNIP/NLRP3 crosstalk has been established in mangiferin, puerarin, and rutin administration that can further mitigate oxidative stress and inflammation in vascular endothelial cells [56,198]. As AGEs induced endothelial dysfunction, salidroside might play a pivotal role by regulating AMPK/NF-κB/NLRP3 signaling [199].

Inhibiting diabetes-related atherosclerosis by raising eNOS levels and decreasing NF-*κ*B and TXNIP expression in response to the priming signal from NLRP3 inflammasome activation, Biejiajian (BJJ), a traditional Chinese medicine, has lately exhibited promising results [12]. The LOX-1/NLRP3 pathway recently emerged as an important mediator in inflammasome activation upon the use of astragaloside IV (AS-IV), the active component of *Astragalus membranaceus* (Fisch.) Bge., an antioxidant treatment [200,201]. 

The aforementioned natural elements function essentially as antioxidants. In part, this can account for their function as NLRP3 inflammasome inhibitors by reducing ROS generation. However, they also provide new insights into the complex interaction between inflammation and oxidation and suggest potential therapeutic options for vascular problems [56]. Current studies are insufficient to completely explain the therapeutic effects of these substances, and further research is necessary to better understand their inflammasome-inhibiting potential.

## 5. Discussion

Although inflammatory biomarkers (hsCRP, IL-6, IL-1, TNF-, MCP-1), endothelial biomarkers (ICAM-1, VCAM-1) and selectins are established in both human and experimental studies as routinely detected in retrieved blood samples, we are still bound by particular detection methods when it comes to quantifying inflammation as a target. While in preclinical studies these changes have been detected in samples of peripheral blood monocytes, macrophages, aortic endothelial cells from mouse tissues and cell cultures using a variety of staining methods and tests (e.g., Western blotting, real-time polymerase chain reaction), the assessment of the therapeutic–inflammatory response is not clearly delineated in most human clinical studies. This underlines the necessity of designing a specific test for sensing and assessing anti-inflammatory therapy that can be implemented in clinical practice in conjunction with a more extensive panel of inflammatory biomarkers.

We need to emphasize some of the contrasts between our research and some of the most important reviews that have been conducted in this topic over the last three years after thorough research. For instance, in their work, Silvis et al. [31] focused on the mechanisms and the involvement of NLRP3 inflammasome strictly in coronary artery disease and acute myocardial infarction with a brief mention of therapy options mentioning only clinical immunotherapy trials. Jiang et al. [43], in their review, discussed in detail the NLRP3 inflammasome components and their pathophysiological implications in atherosclerosis, with a focus only on direct and indirect NLRP3 inflammasome inhibitors and setting aside natural components which could prove an inexpensive source of treatment in the future. Sharma B.R. et al. [38] reviewed the mechanisms of NLRP3 regarding chronic inflammation in atherosclerosis and cancer, without any mention of treatment options, while Sharma A. et al. [132] raised awareness about NLRP3 inflammasome involvement in diabetic atherosclerosis in relation to only one therapeutic option. In contrast, Burger et al. [202] pointed out the role of NLRP3 inflammasome in plaque destabilization and only briefly covered the treatment potential of colchicine. Takahashi et al. [6] covered NLRP3 inflammasome involvement in atherosclerosis in addition to the vascular pathology of different causes (aortic aneurysm, Kawasaki disease). They discuss known clinical trials (e.g., CANTOS, COLCOT) with a concise overview of preclinical studies. Zeng et al. [203] discussed the implications and therapeutic possibilities of pyroptosis but concerning only one aspect of the myriad ramifications of the NLRP3 inflammasome in the pathophysiology of atherosclerosis; they only discuss one therapeutic product.

While pharmacological inhibitors of NLRP3 inflammasome and natural compounds have shown promise in animal models, there are limitations to their use in clinical practice. The lack of comprehensive studies on a larger scale in humans and high-quality research prevents the Food and Drug Administration (FDA) and medical experts from recommending the use of numerous molecules in the form of affordable dietary supplements that contain mainly the natural compounds presented above for the prevention and treatment of atherosclerosis. Furthermore, these agents may have limited bioavailability and can be metabolized quickly, making them less effective in vivo. For example, some inhibitors have low specificity and can also affect other inflammasomes, leading to potential off-target effects. Inhibiting only IL-1 and IL-8 may not fully address the complex inflammatory response downstream of NLRP3 activation, as its activation leads to the production of various cytokines and chemokines that contribute to inflammation, including IL-1β, IL-18, IL-6, TNF-α and IL-8. Additionally, the multiple downstream mediators inhibition treatments targeting IL-1 and IL-18 do not have NLRP3 specificity and can also inhibit the normal immune response pathways. Additionally, the long-term use of interleukin inhibitors is well known for their immune-related adverse events, including low blood pressure, nausea, vomiting, liver enzyme elevations, diarrhea and increased rates of infection. While NLRP3 is a key regulator of inflammasome activation, there is increasing evidence that other inflammasome sensors, such as NLRC4 and AIM2, also play a role in atherosclerosis. Therefore, targeting NLRP3 alone may not be sufficient to fully modulate the inflammasome response in atherosclerosis. Combining NLRP3 inhibitors with other agents that target other aspects of atherosclerosis., such as lipid metabolism or immune cell recruitment, may provide synergistic effects and enhance therapeutic efficacy. Advances in drug discovery and development may lead to identifying new, more specific and potent NLRP3 inhibitors with improved bioavailability and pharmacokinetic properties, potentially resulting in fewer adverse effects. While NLRP3 inflammasome activation is a promising therapeutic target for the treatment of atherosclerosis (Figure 3), and most of its beneficial effects are seen in experimental studies, developing other NLRP3-modifying agents is necessary to address the limitations of current agents. It is also necessary to account for NLRP3-independent mechanisms and explore potential combination therapies. 

As noted, we conclude that our updated review provides scientific data not only about the pathophysiology underlying atherosclerosis, which involves the complex relationship between inflammation, hyperglycemia and oxidative stress and the role of NRLP3 in these pathways, but also encompasses a wider array of therapy information. In addition, our review outlines both preclinical and clinical findings on NLRP3 inhibitors and includes integrative reviews of other lesser-known NLRP3 inhibitors. The aim of this manuscript is to reveal the vastness of this field, which encompasses still unexplored pathways and molecules with biomarker potency and less addressed potential therapeutic targets. Hopefully, it can be of help for future researchers.

**Figure 3 ijms-24-08162-f003:**
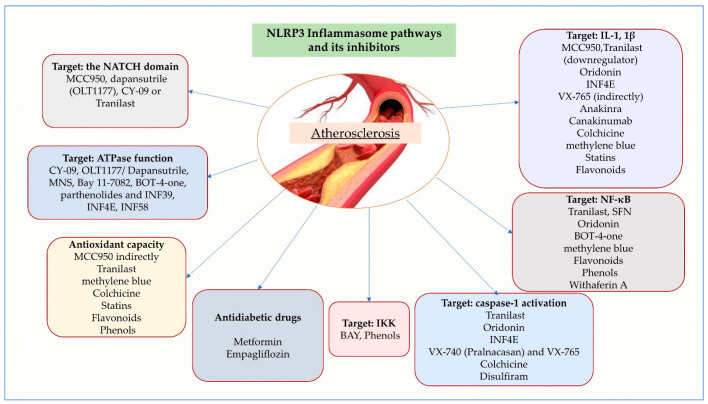
NLRP3 Inflammasome pathways and its inhibitors according to their mechanism of action.

## 6. Conclusions

Unfortunately, as the human population grows, unhealthy lifestyles are increasingly adopted, leading to a rise in the incidence and prevalence of metabolic diseases. All things considered, despite the efforts of bringing new molecules for atherosclerosis treatment and up-to-date management guidelines, atherosclerosis still holds the title of one of the most debilitating diseases when its multiple central and peripheric vascular complications are considered.

For this reason, over the last decades, the scientific community has directed its attention to the supplementary investigation of the pathogenesis behind this disease, with more concentration on inflammasome pathway research. A plethora of evidence noted the role of NLRP3 inflammasome in endothelial dysfunction and atherosclerosis. Human and preclinical studies predominantly displayed the beneficial effects of direct or indirect NLRPR signaling/assembly/activation phase inhibitors. 

A better understanding through translational research could enlighten the full role and potential of inflammasomes in CVD disease. Thus, perhaps the discovery of clinically approved novel pharmacological molecular inhibitors of NLRP3, along with other adjuvant therapy agents, may bring new therapeutic strategies with a reduced burden of atherosclerosis complications.

## Figures and Tables

**Figure 1 ijms-24-08162-f001:**
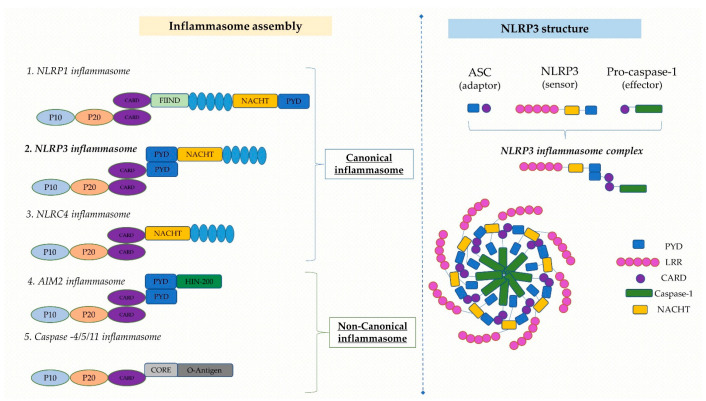
A diagrammatic representation of the assembly of different known inflammasomes and the NLRP3 inflammasome complex structure. Nucleotide-binding and oligomerization domain-like receptor family pyrin domain-containing 3 (NLRP3); NLR Family CARD domain-containing protein 4 (NLRC4); the interferon-inducible protein/absent in melanoma 2 (AIM2); pyrin domain (PYD); leucine-rich-repeat domain (LRR); function-to-find domain (FIIND); caspase recruitment domain (CARD); apoptosis-associated speck-like protein containing a CARD (ASC); caspase-1/ IL-1β converting enzyme (ICE); central nucleotide-binding and oligomerization domain (NACHT/NOD); hematopoietic interferon-inducible nuclear proteins with a 200-amino-acid repeat (HIN-200).

**Figure 2 ijms-24-08162-f002:**
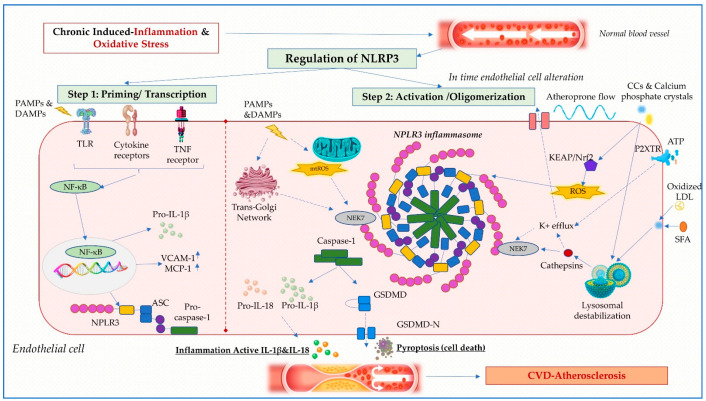
Two step canonical activation of NLRP3 inflammasome-driven downstream events in the arterial endothelial cells. Signal 1 (priming) or ubiquitination phosphorylation, is activated by TLRs or cytokine receptors that recognize and combine the corresponding signals to activate NF-κB at the transcriptional level. The activation signal (Signal 2) is mediated by common intracellular events, such as mitochondrial injury and ROS generation; K+ efflux; lysosome rupture and cathepsin B release; and dispersal of the trans-Golgi network. The NLRP3 inflammasome activates mature caspase-1 that cleaves pro-IL-1b and pro-IL-18 into their active forms. Active caspase-1 also cleaves GSDMD, and its cleaved N-terminus (GSDMD-N) forms subsequently with pyroptosis. ROS = radical oxygen species; nuclear factor kappa-light-chain-enhancer of activated B cells (NF-κB); apoptosis-associated speck-like protein containing a CARD (ASC); P2X purinoreceptor 7 (P2RX7); vascular cell adhesion molecule 1 (VCAM-1); monocyte chemoattractant protein 1 (MCP-1); IL-= in-terleukin-1; IL-18= interleukin-18; never-in-mitosis A-related kinase 7 (NEK7); gasdermin D (GSDMD); terminal domain (GSDMD-N; the up-arrow indicates upregulation.

**Table 2 ijms-24-08162-t002:** Therapeutic pharmacological agents against NLRP3 inflammasome according to their activation phases.

Agent	Study Design	Salient Results	Ref.
Phase I (priming)			
Bay 11-7082	Preclinical (Experimental study)	- Via NF-κB inhibition selectively blocks IKKβ kinase activity with subsequent inhibition of the NLRP3 inflammasome activation; - Inhibited nigericin-induced and MSU-induced caspase-1 activation by the NLRP3 inflammasome.	[148]
NLRP3 oligomerization			
CY-09 (glitazone derivate)	Preclinical (Experimental study)	- It binds to the ATP-binding motif of NLRP3 NACHT domain and inhibits NLRP3 ATPase assembly and activity in macrophages; - Inhibits NLRP3 ATPase activity; - Reverses metabolic effects via NLRP3 inhibition.	[166]
MCC 950	Preclinical (Experimental study)	- Inhibits NLRP3 inflammasome activation by suppressing IL-1β secretion; - It does not affect NLRP1, NLRC4 or AIM2 inflammasomes; - Atheroprotective activity by reducing the size of the plaque; - Reversed the impaired endothelial dysfunction.	[129,130,131]
Dapansutrile	CT (Phase I, randomized controlled trial)	- Inhibits NLRP3-ASC band NLRP3-caspase-1 interaction.	[137]
Tranilast	Approved (Experimental study)	- Reduces ROS, TXNIP expression and directly inhibits xhantine oxidase activity in vitro; - Via binding to the NATCH domain of NLRP3, inhibits assembly and its effects.	[21,36]
Phase II (activation)			
Ang-(1-7)	Preclinical (Experimental study)	- Anti-inflammatory and anti-senescent action through RAAS; - Inhibits IL-1 -induced iNOS expression and NF-κB activation in vascular smooth muscle cells; - Diminishes NLRP3 inflammasome/IL-1 over-activation loop.	[167]
HL2351	CT (Phase I, randomized controlled trial)	-Inhibition of IL-1 function with indirect NLRP3 inflammasome action.	[113]
GSK1070806	CT (Phase II, randomized, placebo-controlled)	- Inhibition of IL-18; - Inhibition of IL-18 did not lead to any improvements in glucose control.	[168]
Rilonacept	Approved (Phase III, double-blind, randomized-withdrawal)	- Inhibition of the IL-1 pathway; - Reduced the activation of endothelial cell NADPH oxidase.	[169]
Canakinumab	Approved (Randomized, double-blind)	- Direct blockade of IL-1 or its receptor; - Antioxidant effects.	[11]
Anakinra	Approved (Randomized, double-blind)	- Modulation of mitochondrial ROS production by activating SOD2.	[11]

Iκβ kinase β (IKKβ); monosodium urate (MSU); clinical trial (CT); the anti-interleukin-18 monoclonal antibody (GSK1070806); renin-angiotensin-aldosterone system (RAAS); superoxide dismutase 2 (SOD2).

## Data Availability

Not applicable.

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
