# Peer review of "Portrayal of NLRP3 Inflammasome in Atherosclerosis: Current Knowledge and Therapeutic Targets"

_ijms, 2023, doi:10.3390/ijms24098162_

Round 1

Reviewer 1 Report

This review focuses on the NLRP3 inflammasome and its complex relationship to atherosclerosis, ranging from basic molecular mechanisms to therapeutic strategies. The contents are presented in detail by many references and are organized to provide a bird's eye view of NLRP3's involvement.

This reviewer provides comments that should be improved upon.

There is missing information in references 8,19,86,149,183.

he abbreviations that appear twice are NLRP3, ATP, TLR, NOX, LDL, CCL2. Again, correct carefully.

1, 2 in Figure 1 left and right, with different abbreviations for NLRP.

5 in Figure 1 left, with different abbreviations for caspases.

The C-terminal domain of AIM-2 is HIN-200, not NIH-200.

The initial letter of “induce” in Figure 2 should be capitalized, with a space before “&DAMP”.

The bottom row, “active”, should also be capitalized.

Different abbreviations for NLRP3 inflammasome.

Put "&" between IL-1b and IL18 in the second column of Table 1.

First row of Table 2, NF-"k "B

Table 2, Ref 169, makes IL-1 an abbreviation.

oX-LDL is unified, with or without hyphen.

Author Response

Dear Reviewer,

Firstly, we want to thank you on behalf of our team for your time and all your observations regarding our manuscript. As recommended, we have revised the abstract, and completed for a better overview of its content. We have taken in consideration all your advices and made the following changes:

  1. This reviewer provides comments that should be improved upon.

Thank you for this observation, as we revised the manuscript we also, enriched the “Discussion” chapter in which we emphasized the current filed quality recent reviews highlights and focus as a different approach, and better pointed out our view and conclusions regarding this domain.   

  1. There is missing information in references 8,19,86,149,183.

We have checked and completed with the missing information.

  1. The abbreviations that appear twice are NLRP3, ATP, TLR, NOX, LDL, CCL2. Again, correct carefully.

We have checked and corrected as suggested to avoid repetition.

  1. 1, 2 in Figure 1 left and right, with different abbreviations for NLRP.

We have checked and corrected as suggested.

  1. 5 in Figure 1 left, with different abbreviations for caspases.

We have checked and corrected as suggested.

  1. The C-terminal domain of AIM-2 is HIN-200, not NIH-200.

We have checked and corrected as suggested.

  1. The initial letter of “induce” in Figure 2 should be capitalized, with a space before “&DAMP”.

We followed your recommendation and corrected.

  1. The bottom row, “active”, should also be capitalized.

We have corrected as suggested.

  1. Different abbreviations for NLRP3 inflammasome.

Thank you for this observation. Also, we identified as much as possible all the repeating    abbreviations, pointed and corrected them.

  1. Put "&" between IL-1b and IL18 in the second column of Table 1.

We followed your recommendation and corrected.

  1. First row of Table 2, NF-"k "B.

We followed your recommendation and corrected.

  1. Table 2, Ref 169, makes IL-1 an abbreviation.

We followed your recommendation and corrected.

  1. oX-LDL is unified, with or without hyphen.

We followed your recommendation and corrected. Thank you once again for all your observations, and recommendation; there were mistakes from out part, therefore to further improve this manuscript, we have corrected any grammatical errors and punctuation mistakes identified through the paper. 

Reviewer 2 Report

Tanase et al provide a review which describes the mechanisms behind NLRP3 activation during Atherosclerosis as well as promising therapeutic targets. The manuscript has a detailed description of NLRP3 activation, with figures and tables that help with the understanding of the topic. Furthermore, the authors also discuss the mechanism through which some treatment might target NLRP3 and improve the outcome. However, the literature has an overwhelming number of reviews with very similar structure, figures and discussion. Some of them are even cited by the authors. Therefore, this review does not bring a novel discussion. Perhaps bringing another approach, still related to NLRP3 and Atherosclerosis, would contribute to highlight the review among others.

The English language needs a moderate revision. In some sections and figures, the abbreviations are wrong.

Author Response

Dear Reviewer,

Firstly, thank you on behalf of our team for your time on peer-reviewing our manuscript. We have taken into consideration all your suggestion, revised the whole manuscript for mistakes and made structural changes, and readjusted the bibliography. Therefore, we believe that with the help of your comments we improved the quality of this paper.

Perhaps bringing another approach, still related to NLRP3 and Atherosclerosis, would contribute to highlight the review among others.

Response:

Upon second reconsideration, we see the point of view of your expert insights therefore to answer to these questions, we have enriched our chapter “5.Discussion” as it fallows:  

- We have also, included data in which we briefly point out the limitation of the actual inflammation biomarkers associated with atherosclerosis and the need for new ones in clinical practice;

- As a different approach, we have made a parallel between the different recent quality reviews in this field in which we highlight their main focus and differences observed compared to our manuscript. Thus, the second paragraph includes the most relevant recent reviews of the last three years; we have researched again using electronic databases, with key words “NLRP3 inflammasomes, atherosclerosis, immunologic pathways, antioxidant therapy”, two of them were not already mentioned in our bibliography (references listed below). Likewise, we mentioned from our point of view how this paper can bring significance to the current project. 

  1. Burger, F.; Baptista, D.; Roth, A.; Da Silva, R.F.; Montecucco, F.; Mach, F.; Brandt, K.J.; Miteva, K. NLRP3 Inflammasome Activation Controls Vascular Smooth Muscle Cells Phenotypic Switch in Atherosclerosis. IJMS 2021, 23, 340, doi:10.3390/ijms23010340.
  2. Zeng, W.; Wu, D.; Sun, Y.; Suo, Y.; Yu, Q.; Zeng, M.; Gao, Q.; Yu, B.; Jiang, X.; Wang, Y. The Selective NLRP3 Inhibitor MCC950 Hinders Atherosclerosis Development by Attenuating Inflammation and Pyroptosis in Macrophages. Sci Rep 2021, 11, 19305, doi:10.1038/s41598-021-98437-3.

- Finally, in the last paragraph of this chapter we point out the limitation of the current known NLRP3 inflammasome inhibitors and other inflammation targets, and why their approval for clinical use is hampered, and other aspects regarding their applications;

Round 2

Reviewer 2 Report

Dear Authors,

I appreciate the additions into the review which make it more distinguish than others. Nevertheless, some grammar revision is still necessary. Moreover, some abbreviations in some figures are still incorrect. 

Moderate editing is necessary.